Vast cryptic diversity in direct-developing frogs Pristimantis (Anura: Strabomantidae): a new subgenus and the description of a new species from the eastern Andes of Ecuador

Loza-Carvajal Keyko D. 1
http://orcid.org/0000-0003-3224-1987 Yánez-Muñoz Mario H. 2
http://orcid.org/0009-0003-6412-3165 Quilumbaquin Walter 1
http://orcid.org/0000-0001-9464-3688 Ortega-Andrade H. Mauricio 1 2 3 mauricio.ortega@ikiam.edu.ec
1 Integrative Biology Laboratory, Universidad Regional Amazónica Ikiam , Tena, Napo , Ecuador
2 Unidad de Investigación, Instituto Nacional de Biodiversidad (INABIO) , Quito, Pichincha , Ecuador
3 Biogeography and Spatial Ecology Research Group, Life Sciences Faculty, Univerisidad Regional Amazónica Ikiam , Tena, Napo , Ecuador
Brygadyrenko Viktor
Electronic publication date: 2025 Dec 17
Publication date: 2025
Volume: 13
Electronic Location ID: e20512
Received 2025 Sep 11; Accepted 2025 Nov 11
Copyright: © 2025 Loza-Carvajal et al.
Copyright year: 2025
Copyright holder: Loza-Carvajal et al.
License: This is an open access article distributed under the terms of the Creative Commons Attribution License, which permits unrestricted use, distribution, reproduction and adaptation in any medium and for any purpose provided that it is properly attributed. For attribution, the original author(s), title, publication source (PeerJ) and either DOI or URL of the article must be cited.
License URL: https://creativecommons.org/licenses/by/4.0/

Keywords: Integrative taxonomy, Cryptic diversity, Pristimantis paganus sp. nov., Pristimantis prolatus group, Pristimantis bicantus group, Cryptomantis gen. nov.

Funding: Secretaría Nacional de Ciencia y Tecnología del Ecuador (Senescyt-ENSAMBLE) #PIC-17-BENS-001 The World Academy of Sciences #16-095 This work was supported by the project “On the quest of the golden fleece in Amazonia: The first herpetological DNA—barcoding expedition to unexplored areas on the Napo watershed, Ecuador” with a grant from the Secretaría Nacional de Ciencia y Tecnología del Ecuador (Senescyt-ENSAMBLE Grant #PIC-17-BENS-001), The World Academy of Sciences (TWAS Grant #16-095) awarded to HMOA). The funders had no role in study design, data collection and analysis, decision to publish, or preparation of the manuscript.

==============================
Pristimantis, a genus of direct-developing frogs within the family Strabomantidae, comprises 617 recognized species, making it the most species-rich genus of vertebrates worldwide. This group include 264 described (43% of the world) species in Ecuador, being one of the countries in the region with the highest rate of species description, greatest diversity and endemism. In this study, we analyze the phylogenetic position and describe a new Pristimantis species from the Colonso Chalupas Biological Reserve and Llanganates National Park in northeastern Ecuador, using phylogenetic, genetic, morphological, and geographic evidence. Additionally, we propose a new subgenus within Pristimantis that includes the Pristimantis prolatus and Pristimantis bicantus species groups. Our results indicate that the new species and related species form a well-supported group with significant genetic divergence based on the 16S rRNA gene (average uncorrected p-distance = 2.8–7.5%), within the Pristimantis bicantus species group. Morphologically, the new species is characterized by a black to dark-gray dorsum and marbled venter (less intense or brown in males), being endemic to the Guacamayos mountain range and the Llanganates region in the northeastern Andean foothills of Ecuador. We emphasize the importance of including topotypic specimens to analyze and compare species groups to delimiting species, like Pristimantis.

Introduction

Amphibians are one of the most diverse components of the vertebrate fauna in Ecuador, with a total of 700 species recorded (Ron, Merino-Viteri & Ortiz, 2024). The genus Pristimantis (Strabomantidae) is a group of terrestrial frogs with direct development, identified as the most diverse land-vertebrate genus in the Neotropics (Waddell et al., 2018). Pristimantis frogs include 264 described (43% of the world) species in Ecuador (Ron, Merino-Viteri & Ortiz, 2024), being one of the countries in the region with the highest rate of species description, greatest diversity and endemism (Hedges, Duellman & Heinicke, 2008; Reyes-Puig & Mancero, 2022). However, it is suspected a vast richness underestimation (Bickford et al., 2007) due to the taxonomic complexity, incomplete phylogenies and high phenotypic variation, within the group. In Ecuador, almost 60% of Ecuadorian species of this genus considered at risk of extinction (Ortega-Andrade et al., 2021).

Over the past two decades, the description of new species of Pristimantis has significantly increased (Brito & Almendáriz, 2018; Brito & Pozo-Zamora, 2013; Reyes-Puig & Yánez-Muñoz, 2012; Yánez-Muñoz et al., 2010). Although taxonomic work based on morphology has notably advanced our understanding of this group (Acevedo, Pallares & Perez, 2014; Acosta-Galvis, 2015; Lehr & Coloma, 2008; Maciel et al., 2012; Reyes-Puig et al., 2015; Reyes-Puig & Yánez-Muñoz, 2012; Yánez-Muñoz, 2014), relying solely on morphological data can overlook cryptic diversity (Lehr & Coloma, 2008; Ortega-Andrade et al., 2015; Páez & Ron, 2019). Recent studies have identified morphological synapomorphies in Brachicephaloidea, but no synapomorphies have been found for phenotypic or phylogenetic species groups within Pristimantis, except for the P. conspicillatus species group (Heinicke, Barrio-Amoros & Hedges, 2015; Ospina-Sarria & Grant, 2021; Ron et al., 2020). Conversely, some historical diagnostic traits, such as dorsolateral folds, coloration, dorsal tubercles, and cranial ridges (Duellman & Lehr, 2009), have shown phenotypic plasticity and homoplasy (Guayasamin et al., 2015).

Integrative taxonomy provides a more comprehensive understanding of species boundaries and relationships by cross-validating different types of data and complementary perspectives (phylogeography, comparative, morphology, population genetics, ecology, development, behaviour, etc.) (Domínguez-Domínguez & Vázquez-Domínguez, 2009). From a taxonomic perspective, the description and delimitation of species is important since it provides information to estimate the conservation status of genealogical lineages (Dayrat, 2005; De Queiroz, 2003; De Queiroz, 2007), especially of morphologically cryptic groups (Ortega-Andrade et al., 2021).

A group of frogs related to Pristimantis prolatus (Lynch & Duellman, 1980) and P. suetus (Lynch & Rueda-Almonacid, 1998), found on the eastern slopes of the Andes in Ecuador, was reported by Ortega, Brito & Ron (2022) to form a monophyletic group. Several species in this clade, including Pristimantis bicantus (Guayasamin & Funk, 2009), P. nelsongalloi (Valencia et al., 2019) and P. sacharuna (Reyes-Puig et al., 2015), were phylogenetically linked for the first time within a complex group that also includes at least three putative undescribed species. These species reveal hidden cryptic diversity along the montane forests of the eastern Andean slopes in Ecuador (Ortega, Brito & Ron, 2022).

The Llanganates National Park and the Colonso Chalupas Biological Reserve are areas vastly unexplored in the northeastern Andes of Ecuador. Since 2016, research programs from Instituto Nacional de Biodiversidad (INABIO) and Universidad Regional Amazónica Ikiam, are dedicated to document the biodiversity from both conservation areas, where cryptic diversity in Pristimantis frogs has been recorded (Guayasamin & Funk, 2009; Ortega, Brito & Ron, 2022). The conservation of these areas is important since they harbor a unique diversity of species belonging to sensitive ecosystems, including Páramo highlands and cloud forests. In contrast, some threats on biodiversity has been registered for those areas, related with illegal mining, habitat loss and forest exploitation (Paz Cardona, 2022).

In this work, we describe a new subgenus of Pristimantis and a new species of Pristimantis from Colonso Chalupas Biological Reserve and Llanganates National Park, northeastern Ecuador, based on molecular phylogenetic, morphological and geographic lines of evidence.

Materials and Methods

The electronic version of this article in Portable Document Format (PDF) will represent a published work according to the International Commission on Zoological Nomenclature (ICZN), and hence the new names contained in the electronic version are effectively published under that Code from the electronic edition alone. This published work and the nomenclatural acts it contains have been registered in ZooBank, the online registration system for the ICZN. The ZooBank LSIDs (Life Science Identifiers) can be resolved and the associated information viewed through any standard web browser by appending the LSID to the prefix http://zoobank.org/. The LSID for this publication is: LSID urn:lsid:zoobank.org:pub:11445DCE-6CC1-4BA6-9ACC-1B070E79AA24. The online version of this work is archived and available from the following digital repositories: PeerJ, PubMed Central SCIE and CLOCKSS.

Ethics statement

Specimens and tissue samples obtained following technical protocols proposed by Heyer et al. (1994). Type specimens were deposited at the Instituto Nacional de Biodiversidad INABIO, Quito, Ecuador (DHMECN), under permits MAE-DNB-CM-2016-0045, 2017-0062 and 2019-0120 issued by the Ministry of Environment, Water and Ecological Transition of Ecuador.

Fieldwork

Collections were carried out in 2016 (0.93527°S, 77.92683°W, 2206 m a.s.l.), 2017, and 2021 (0.93829°S, 77.94833°W, 2157 m a.s.l.) at the Colonso Chalupas Biological Reserve, Napo Province, Ecuador, and at Cerro de Abitagua (1.36217°S, 78.10865°W, 2200 m a.s.l.) in Llanganates National Park, Pastaza Province, Ecuador, during the nights of December 12 to 24, 2021. Maps were created using the Digital Elevation Model (DEM) developed by the Ecuadorian Government through the Ministry of Agriculture’s geospatial system, SIGTIERRAS, available at: http://geoportal.agricultura.gob.ec:8090/sinat_web_descarga/login.

Specimens were photographed alive, euthanized with 2% lidocaine by immersion (5–10 min), a sample of liver tissue was extracted, labeled with the respective field code and stored in 96% ethanol. A 10% solution of formalin was used to fix specimens left by 24 h after capture and then preserved in 70% ethanol.

Taxon sampling

We follow Ortega-Andrade et al. (2015) and their protocol outlines steps for re-evaluating the taxonomy of a species complex: (1) Selection of specimens for molecular analysis and review of the type series of taxa within the focus group, (2) conducting comparative molecular analyses (e.g., phylogenetics, genetic distances), and (3) performing comparative analyses of various lines of evidence to delimit species based on qualitative-quantitative morphological characters and biogeography. In general, specimens selected for morphological analyses and taxonomic descriptions were chosen after phylogenetic analyses (step two of the integrative protocol) and based on taxonomically assessed diagnostic characters for each species (step three of the protocol).

Molecular assay for tissue samples

Genomic DNA extraction was performed using the Wizard® Genomic DNA Purification kit (Promega, Madison, WI, USA), with 5–10 mg of liver or muscle tissue (see Table S1), following the manufacturer’s protocol. The concentration and purity of genomic DNA were determined with the NanoDrop™ One/Onec Microvolume UV-Vis spectrophotometer (Thermo Scientific, Waltham, USA). We amplified DNA samples by Polymerase Chain Reaction (PCR) using three mitochondrial genes: 16S rRNA, 12S rRNA, and cytochrome oxidase subunit 1 (COI) and one nuclear gene recombination activating Gene 1 (RAG1). The final reaction volume was 15 µl and consisted of 2× DreamTaq Green (Invitrogen, Carlsbad, CA, USA), nuclease-free water, 50 ng/µL bovine serum albumin (BSA; Invitrogen, Carlsbad, CA, USA), 0.2 µM forward primer, 0.2 µM reverse primer and, 50 ng/µl of genomic DNA. PCR amplification was performed on 9 specimens according to the protocols of Pinto-Sánchez et al. (2012) and Sanger sequencing by Macrogen (Seoul, South Korea).

A second round of PCR amplification was performed on 20 specimens for sequencing using Oxford Nanopore Technology (ONT) at the Laboratory of Molecular Biology and Biochemistry, Universidad Regional Amazónica Ikiam (Table S1). For short amplicons, the following primers were used: 16sSar-L and 16sSbr-H for 16S rRNA; 12sL4E and 12sH10 for 12S rRNA; dgLCO-1490 and dgHCO-2198 for COI; and R182 and R270 for RAG1. Primers 12sL4E and 16sSbr were used to generate long fragments (approx. 2,400 bp) of the 12S rRNA (868 bp) and 16S rRNA (1,400 bp) genes (PCR conditions and primers are detailed in Table S2).

We performed two library preparations for ONT sequencing: the first was for short amplicons using the Ligation Sequencing Amplicons—PCR Barcoding protocol (SQK-LSK109 with EXP-PBC096, ONT, Oxford, UK), and the second was for long fragments, using the Ligation Sequencing Amplicon—Native Barcoding—24 V14 protocol (SQK-NBD114.24, ONT, Oxford, UK). Both preparations followed the manufacturer’s guidelines. The library preparations were purified with Agencourt 1.8× AMPure XP beads (Beckman Coulter, Brea, CA, USA) and quantified using the Qubit 4 Fluorometer kit (Invitrogen, Carlsbad, CA, USA).

We used two Flongle Flow Cells for sequencing: the R9 version (FLO-FLG001) for short fragments and the R10 version (FLO-FLG114) for long fragments. Prior to final library loading, the Flongle flow cells were conditioned with a mixture of 117 µl Flow Cell Flush (FCF) and 3 µl Flow Cell Tether (FCT) from the Flow Cell Priming Kit (EXP-FLP002). The sequencing mix had a final volume of 30 µl, consisting of 15 µl sequencing buffer (SB), 10 µl library beads (LIB), and 25 fmol of DNA library in a total volume of 5 µl.

Sequencing was performed on a MinION Mk1C device (ONT, Oxford, UK). The sequencer used MinKNOW v23.07.12 and Guppy v7.1.4 software to manage the processes during the run and perform basecalling and demultiplexing of the reads based on their barcodes. The sequencing run lasted 15 to 20 h until all available pores were exhausted. The raw data were processed using the Porechop v0.2.4 (https://github.com/rrwick/Porechop) and NGSpeciesID v0.3.0 (https://github.com/ksahlin/NGSpeciesID) tools (Egeter et al., 2022; Quilumbaquin et al., 2023). Nanopore sequences were edited and aligned in Geneious Prime v2024.0.1 (Drummond et al., 2024).

Phylogenetic analysis

We generated a total of 81 consensus sequences (10 for 16S, 10 for 12S, 17 for 12S&16S, 22 for COI, and 22 for RAG1) from 29 individuals (see Table S2). We performed an initial phylogenetic reconstruction to determine the position of the complex group studied within Pristimantis. This analysis utilized a matrix that included 1,155 taxa terminals and 2,981 characters (see Supplemental Material S1 for the matrix).

A second phylogenetic reconstruction, focusing on the complex group relationships, was developed using new sequences and genetic data deposited in GenBank. For the first time, we incorporated new sequences from Pristimantis tungurahua (Reyes-Puig et al., 2010), P. marcoreyesi (Reyes-Puig et al., 2014), P. ganonotus (Duellman & Lynch, 1988), P. burtoniorum (Reyes-Puig et al., 2022), and additional sequences from P. bicantus and P. prolatus. DNA sequences were visually inspected and re-aligned to avoid non-homologous characters, following initial alignment with the MAFFT v7.017 (Multiple Sequence Alignment) algorithm (Katoh, Rozewicki & Yamada, 2017), using default settings. To minimize unaligned characters in the concatenated matrix, sequence gaps were removed using Geneious Prime v2022.0.1 software (Kearse et al., 2012). Craugastor longirostris, and Niceforonia elassodiscus were chosen as outgroups (Duellman & Lehr, 2009) in both phylogenetic analyses. The matrix, which included 64 taxa terminals and 3,371 bp (Supplemental Material S3), was concatenated and analyzed using Mesquite v3.61 (Maddison & Maddison, 2015).

Phylogenetic trees were estimated to be using a maximum likelihood (ML) algorithm in IQTree v2.2.0 web-server (Nguyen et al., 2015). Nucleotide substitution models were estimated with ModelFinder (Kalyaanamoorthy et al., 2017) implemented in the IQTree server, for a total of eight partitions, one for each non-coding gene (12S and 16S) and one for each codon of the coding genes (COI and RAG1). Support for node values was estimated using the Shimodaira-Hasegawa-like approximate likelihood ratio test (SH-aLRT) (Guindon et al., 2010) with 1,000 replicates and 1,000 Ultrafast Bootstrap (UFBoot) (Minh, Nguyen & von Haeseler, 2013). The phylogeny was visualized and edited using FigTree v1.4.2 (Rambaut, 2014) and Inkscape v1.3.2. https://inkscape.org/es/. Uncorrected pairwise genetic distances (p-distances) between the new species and other species in the clade were estimated using 16S rRNA data from the aligned matrix in Geneious Prime v2024.0.7 (https://www.geneious.com). Sequences that did not show homology were excluded from the analysis.

Definition of candidate species

We follow the concept defined by Vieites et al. (2009) to recognize Unconfirmed Candidate Species (CS) in this study. Specimens were characterized by a detectable genetic differentiation of more than 3%, or 2% according to several studies on Pristimantis (Franco-Mena et al., 2023; Ortega-Andrade et al., 2015; Ortega, Brito & Ron, 2022). However, for some taxa, data could be deficient or incomplete for morphology, ecology, or distribution due to unavailability of voucher specimens or immature state of vouchers (Vieites et al., 2009), that are commonly reported for this group (Ortega, Brito & Ron, 2022; Páez & Ron, 2019; Ron et al., 2020).

Morphological analysis

The collected material was examined and compared with specimens deposited in the Herpetological Collection (DHMECN) from the Instituto Nacional de Biodiversidad, Quito (INABIO) and KU Herpetology Collection, University of Kansas, USA. The taxonomic terminology and diagnosis follows the proposal of Duellman & Lehr (2009). The presence of the hyperdistal tubercle in the new species follows the codification by Ron et al. (2020). Morphometric measurements were taken with an electronic caliper (accuracy ± 0.01 mm). The following morphological measurements were taken (Duellman & Lehr, 2009). Codification of subarticular tubercles follow the classification of Ron et al. (2020). Sex, maturity of specimens and reproductive condition were delimited by identification of vocal slits, size and through direct observation of gonads by dorsolateral sectioning. Color in life was determined based on photographs taken in the field. Sexual dimorphism is analyzed with a Principal Component Analysis to detect the variables which explain the variation among sexes. Statistical analyses (PCA, Mann-Whitney U test) to compare morphometric measurements were performed in PAST software V4 (Hammer et al., 2001).

Results

Phylogenetic relationships

Our phylogeny identifies a high-supported monophyletic (bootstrap = 100) clade within Pristimantis (Fig. 1), with two subclades with high support values (bootstrap = 99). The major clade is described herein as Cryptomantis subgenus nov. (See Systematic accounts, for taxonomy description). The first subclade (A) is taxonomically assigned to the Pristimantis prolatus species group, which include four species (P. ganonotus, P. burtoniorum, P. prolatus, P. c.sp. 1), whereas the second subclade (B), is composed of 10 species (P. c.sp. 2, P. sacharuna, P. c.sp. 3, P. c.sp. 4, P. c.sp. 5, P. nelsongalloi, P. bicantus, P. tungurahua, P. marcoreyesi, and Pristimantis paganus sp. nov.), being assigned to the Pristimantis bicantus species group (Fig. 1). The phylogenetic position for the holotype of P. burtoniorum and the topotypes of P. tungurahua, P. marcoreyesi and P. prolatus are reported for the first time.

Figure 1 (A) Phylogenetic relationships of the genus Pristimantis, detailed in Fig. S1. (B) Maximum likelihood tree of the subgenus Cryptomantis, whereas the position of Pristimantis paganus sp. nov. is shown in blue.

SH-aLRT support (%)/ultrafast bootstrap support (%) values are shown for each branch. The voucher specimen number for each terminal is shown before the species name. Topotypes and localities of collection are shown associated to each clade, in colors. Unconfirmed candidate species are shown in red.

We identified an undescribed species of Pristimantis (P. paganus sp. nov.; see Systematics account for description) grouped in a high-supported clade (bootstrap: 98%) together with its sister species, Pristimantis marcoreyesi, and P. c.sp. 2; all of them included in the Pristimantis bicantus species group.

The 16s rRNA uncorrected p-distance between P. paganus sp. nov. and related species within the P. bicantus species group range from 0.024–0.11 (0.06 ± 0.018 Standard deviation [SD]); distances with P. marcoreyesi 0.024–0.031 (0.028 ± 0.002 SD) with P. csp. 2 (0.03–0.062 (0.04 ± 0.01)) (Table 1). The genetic distances compared with other species range from 0.024–0.17 (0.08 ± 0.04) (Table S4).

Table 1 Mean interspecific genetic p-distances.

Values below the diagonal correspond to p-distances in the 16S rRNA gene between P. paganus sp. nov. and its closest nominal species. The minimum, maximum and standard deviation are shown above the diagonal.

	P. paganus sp. nov	P. bicantus	P. marcoreyesi	P. nelsongalloi	P. sacharuna	P. tungurahua	
P. paganus sp. nov		0.031–0.08 (±0.01)	0.024–0.031 (±0.002)	0.036–0.105 (±0.015)	0.032–0.077 (±0.013)	0.046–0.076 (±0.01)	
P. bicantus	6.5%		0.051–0.086 (±0.01)	0.034–0.11 (±0.015)	0.047–0.051 (±0.002)	0.025–0.072 (±0.011)	
P. marcoreyesi	2.8%	7.2%		0.074–0.103 (±0.01)	0.058	0.044–0.091 (±0.015)	
P. nelsongalloi	7.5%	6.4%	8.4%		0.039–0.056 (±0.01)	0.029–0.093 (±0.015)	
P. sacharuna	4.5%	4.8%	5.8%	4.5%		0.047	
P. tungurahua	5.2%	6.2%	6.9%	6.6%	4.7%		

Systematic account

Cryptomantis subgenus nov.

LSID urn:lsid:zoobank.org:act:AD83BF8F-72C0-4453-ABA8-705C2F258944.

Type species. Pristimantis prolatus (Lynch & Duellman, 1980).

Definition. This clade is strongly supported by phylogenetic evidence (Fig. 1). Morphological synapomorphies are unknown. Members of this clade are characterized by: (1) head narrower than body, (2) tympanic membrane and tympanic annulus present, (3) cranial crests absent, (4) dentigerous processes of vomer present (except for P. ganonotus), (5) small to large, distinct tubercles on heel present (except for P. bicantus), (6) “S” condition of adductor muscle, (7) terminal discs of digits truncate, expanded, slightly expanded or lanceolate, with circummarginal grooves, (8) Finger I shorter than second, (9) Toe V longer than Finger III, (10) subarticular tubercles, present, prominent; hyperdistal subarticular tubercle, present (11) texture of dorsum variable, finely shagreen to tuberculate, usually with distinctive dorsolateral dermal folds and tubercles, (12) texture of venter areolate, (13) SVL range of adult individuals from 12 to 22.2 mm in males and 12 to 30.8 mm in females.

Diversity (Figs. 1, 2) The subgenus Cryptomantis gen. nov. include nine described species: P. bicantus (Guayasamin & Funk, 2009), P. burtoniorum (Reyes-Puig et al., 2022), P. ganonotus (Duellman & Lynch, 1988), P. marcoreyesi (Reyes-Puig et al., 2014), P. nelsongalloi (Valencia et al., 2019), P. paganus sp. nov., P. prolatus (Lynch & Duellman, 1980), P. sacharuna (Reyes-Puig et al., 2015) and P. tungurahua (Reyes-Puig et al., 2010).

Figure 2 Representative species of the subgenus Cryptomantis.

Subclade A-Pristimantis prolatus species group: (A) Pristimantis ganonotus DHMECN 16961, Colonso Chalupas Reserve, Napo; (B) P. burtoniorum DHMECN 14479, Paratype, Mayordomo Reserve, Tungurahua; (C) P. prolatus DHMECN 11564, Topotype, Río El Reventador, Sucumbíos; (D) P. c.sp. 1, DHMECN 15674, Colonso Chalupas Reserve, Napo. Subclade B-Pristimantis bicantus species group: (E) P. c.sp. 2, QCAZ 70020, Llanganates National Park, Napo; (F) P. marcoreyesi, DHMECN 13833, Tungurahua volcano, Tungurahua; (G) P. paganus sp. nov. DHMECN 15606, Colonso Chalupas Reserve, Napo; (H) P. tungurahua, DHMECN 14428, Vizcaya Reserve, Tungurahua; (I) P. sacharuna, QCAZ 52496, Zúñag Reserve, Tungurahua; (J) P. c.sp. 3, QCAZ 51553, San Antonio de Juval, Cañar; (K) P. nelsongalloi, DHMECN 5223, Zúñag Reserve, Tungurahua; (L) P. bicantus, DHMECN 12359, El Reventador, Sucumbíos; (M) P. c.sp. 5, QCAZ 52489, Sangay National Park, Morona Santiago. Photo credit: Gustavo Pazmiño, Bioweb (E); Santiago Ron, Bioweb (I), Bioweb (J, M); Keyko D. Loza, Carvajal (A); Mario H. Yánez Muñoz (B, C, F, H, K, L); H. Mauricio Ortega-Andrade (D, G).

Distribution. The distribution range of the subgenus is restricted to the montane forests of the eastern foothills and adjacent to Amazonian Mountain ranges of Ecuador, ranging from the north, the upper basin of the Napo and Pastaza, including the headwaters of the Pastaza River, to the south, in the headwaters of the Nangaritza River.

Etymology. The name Cryptomantis is derived from the Greek kryptos (hidden) and mantis (tree frog), in reference to the remarkable cryptic diversity observed within the clade. Historically, species belonging to this group have often been misidentified as members of other species groups, and their true diversity has been underestimated.

Comments. Within the subgenus Cryptomantis, two well-supported subclades are identified: the Pristimantis (Cryptomantis) prolatus species group (P. burtoniorum, P. ganonotus, P. prolatus, P. c.sp. 1) and the Pristimantis (Cryptomantis) bicantus species group (P. bicantus, P. marcoreyesi, P. nelsongalloi, P. sacharuna, P. tungurahua, P. c.sp. 2, , P. c.sp. 3, P. c.sp. 4, P. c.sp. 5, and Pristimantis paganus sp. nov. (Fig. 1).

Pristimantis (Cryptomantis) prolatus species group new taxon

Definition: Small-sized frogs with relatively long limbs; SVLs ranges from 13.7 to 18.4 mm in males and 20.8 to 27 mm in females. Dorsolateral dermal folds present (except in P. burtoniorum and P. ganonotus); head width 32.5–39.6% of SVL. The tympanic membrane and annulus are distinctive. Dorsum is finely shagreen to tuberculate, venter areolate. The toes have no interdigital membranes, and the Toe V is much longer than Toe III. Lateral fringes are weak or absent.

Diversity. This group include three described species: Pristimantis burtoniorum (Reyes-Puig et al., 2022), P. ganonotus (Duellman & Lynch, 1988), Pristimantis prolatus (Lynch & Duellman, 1980) and one Unconfirmed Candidate Species (Figs. 1, 2).

Distribution. The distribution range of the group extends from the north, encompassing the montane forests of the eastern foothills of the Andes in the upper Napo basin, including the headwaters of the Pastaza River, to the south, reaching the headwaters of the Nangaritza River in Zamora Chinchipe.

Comments. The only available sequence of Pristimantis prolatus (KU 177433) (Lynch & Duellman, 1980), collected near the type locality, has been shown to represent a lineage independent from other monophyletic groups within Pristimantis (Hedges, Duellman & Heinicke, 2008; Padial, Grant & Frost, 2014). The recent publication by Ortega, Brito & Ron (2022), provided more sequences from P. prolatus (sensu lato), excluding the sequence KU 177433 (Hedges, Duellman & Heinicke, 2008), shown that are related, with low support values, to P. suetus (Lynch & Rueda-Almonacid, 1998), and a clade conformed by the Pristimantis bicantus species group. Our phylogeny contributes topotypic sequences of Pristimantis prolatus (sensu stricto), resulting in the identification of the Unconfirmed Candidate Species 1 and its relationship with P. burtoniorum in the group (Reyes-Puig et al., 2022). Unlike (Ortega, Brito & Ron, 2022), we obtained a high support for the clade formed for the species corresponding to the P. prolatus and P. bicantus species groups. For Pristimantis suetus (MHUA 4404), a 502 bp fragment of 16S rRNA places it within a clade that also includes P. platychilus (Lynch, 1996) and P. permixtus (Lynch, Ruiz-Carranza & Ardila-Robayo, 1994) which are from western and north-central Colombia, respectively (see Fig. S1).

Pristimantis (Cryptomantis) bicantus species group new taxon

Definition. Small-sized frogs with relatively long limbs; SVL ranges from 11.8 to 22.2 mm in males and 12 to 30.8 mm. Dorsolateral dermal folds are present (except in P. bicantus), but weak in P. nelsongalloi and P. paganus sp. nov. Head width 37.7–42.4% of SVL. The tympanic membrane and annulus are distinctive. Dorsum finely shagreen with tubercles, venter areolate. The toes have no interdigital membranes, and the Toe V is much longer than Toe III. Lateral fringes are weak or absent. Vocal slits and nuptial pads present or absent.

Diversity. This group include six described species: Pristimantis bicantus (Guayasamin & Funk, 2009), P. marcoreyesi (Reyes-Puig et al., 2014), P. nelsongalloi (Valencia et al., 2019), P. paganus sp. nov, P. sacharuna (Reyes-Puig et al., 2015), P. tungurahua (Reyes-Puig et al., 2010), and four Unconfirmed Candidate Species (Figs. 1, 2).

Distribution. The distribution range of the group is from the north, in the montane forests of the eastern foothills of the Andes in the upper Napo basin, including the headwaters of the Pastaza River, to the south, in the headwaters of the Nangaritza River in Zamora Chinchipe.

Comments. The recent publication by Ortega, Brito & Ron (2022) provided the first sequences for P. bicantus (sensu lato), P. nelsongalloi (sensu stricto), P. sacharuna (sensu stricto), and a set of three unconfirmed candidate species. Their phylogenetic analysis showed that these specimens were strongly supported as part of a clade within the Pristimantis bicantus species group, which was left unnamed. Our phylogeny includes topotypic sequences of P. bicantus (sensu stricto) from Yanayacu Station, Napo, and positions them with a set of sister species (P. tungurahua P. nelsongalloi and P. sacharuna) and three unconfirmed candidate species (P. c.sp. 3 to P. c.sp. 5).

New species

Pristimantis paganus sp. nov.

LSID:urn:lsid:zoobank.org:act:D532641A-414E-4580-A506-FDAEFDA34B0E

Suggested common English name: Amazonian pagan rainfrog

Common name in Spanish: Cutín pagano Amazónico.

Holotype (Figs 3–5). DHMECN 16810, adult female, from the Colonso Chalupas Biological Reserve, camp 3, Alto Tena, San Juan de Muyuna, Napo province, Ecuador, (0.93158°S, 77.95659°W), 2,495 m, collected on 24 November 2021, by H. Mauricio Ortega-Andrade, Keyko Loza, Mario H. Yánez-Muñoz, Miguel Urgilés, Jorge Brito, and Mauricio Herrera.

Figure 3 Pristimantis paganus sp. nov.

Preserved holotype DHMECN 16810, adult female, SVL = 27.2 mm. (A) Dorsal view; (B) ventral view; (C) lateral view. Photographs by Mario H. Yánez-Muñoz.

Figure 4 Pristimantis paganus sp. nov.

Preserved holotype DHMECN 16810, adult female, SVL = 27.2 mm. (A) Palmar surface; (B) plantar surface; (C) dorsal view of the head; (D) lateral view of the head. Photographs by Mario H. Yánez Muñoz.

Figure 5 Pristimantis paganus sp. nov.

Holotype in life DHMECN 16810, adult female, SVL = 27.2 mm. (A) Dorsal view; (B) frontal view; (C) lateral view; (D) ventral view. Photographs by Mario H. Yánez Muñoz.

Paratypes (Figs 6, 7). A total of 6 females, 13 males and 3 juveniles. Adult female collected at the same locality as holotype: DHMECN 16811, collected on 25 November 2021; Males collected at the same locality as holotype DHMECN 16812–4, collected on 25 November 2021 by Mario H. Yánez Muñoz, Miguel Urgilés, Jorge Brito, H. Mauricio Ortega-Andrade, Mauricio Herrera and Keyko Loza; Adult males: DHMECN 15592–3 (0.93824°S, 77.94553°W, 2,200 m) collected by Jimmy Velasteguí, Grace Reyes, Michelle Guachamin and H. Mauricio Ortega-Andrade on 19 December 2016; DHMECN 15594 (0.93846°S, 77.94770°W, 2,230 m) collected on 20 December 2016 by Jimmy Velasteguí, Grace Reyes, Michelle Guachamin, and H. Mauricio Ortega-Andrade; DHMECN 15595–15596, DHMECN 15598–15599 (0.93874°S, 77.94742°W, 2,221 m) collected by Jimmy Velasteguí, Grace Reyes, Michelle Guachamin and H. Mauricio Ortega-Andrade on 18 June 2017; DHMEC 15600–15603 (0.93685°S, 77.94993°W, 2,216 m) collected by H. Mauricio Ortega-Andrade on 19 June 2017; DHMECN 15605 (0.93829°S, 77.94833°W, 2,157 m) collected by Miguel Gómez, Grace Reyes, Salomón Ramírez and H. Mauricio Ortega-Andrade on 16 November 2018; Juveniles: DHMECN 15596, DHMECN 15599 (0.93844°S, 77.94641°W, 2,223 m), DHMECN 15603 (0.93844°S, 77.94641°W, 2,223 m) collected by H. Mauricio Ortega-Andrade on 18 June 2017; Females: DHMECN 15598 (0.93844°S, 77.94641°W, 2,223 m) 18 June 2017, DHMECN 15600 (0.93685°S, 77.94993°W, 2,216 m) 19 June 2017, DHMECN 15606 (0.93817°S, 77.94840°W, 2,200 m) 16 November 2018, collected by H. Mauricio Ortega-Andrade, Grace Reyes, Miguel Gómez and Salomón Ramírez.

Figure 6 Variation of dorsal and ventral coloration in preserved specimens corresponding to the type series of Pristimantis paganus sp. nov.

From left to right: (A) DHMECN 16810, female, holotype, SVL = 27.2 mm; (B) DHMECN 15606, female, paratype, SVL = 29.8 mm; (C) DHMECN 16811, female, paratype, SVL = 24.8 mm; (D) DHMECN 15605, male, paratype, SVL = 21.4 mm; (E) DHMECN 16812, male, paratype, SVL = 20.2 mm; (F) DHMECN 16814, male, paratype, SVL = 19.1 mm; (G) DHMECN 16813, male, paratype, SVL = 17.2 mm. Photographs by Mario H. Yánez Muñoz (A, C, E–G), and H. Mauricio Ortega-Andrade (B, D).

Figure 7 Dorso-lateral and ventral coloration of Pristimantis paganus sp. nov. in life.

(A) DHMECN 15598, female, paratype, SVL = 29.2 mm; (B) DHMECN 15600, female, paratype, SVL = 29.8 mm; (C) DHMECN 15601, male, paratype, SVL = 20.1 mm; (D) DHMECN 16813, male, paratype, SVL = 17.2 mm. Photograph: H. Mauricio Ortega-Andrade (A–C), Mario H. Yánez Muñoz (D).

Referred specimens. DHMECN 17223 male collected on 01 December 2021, DHMECN 17221 on 28 November 2021, DHMECN 17222 on 29 November 2021 collected by Zane Libke, Eli Bieri, Kira Miller, Sara Dykman, Henry Sanchez, Vicente Sanchez, Jordi Salazar, Kira Miller, Sara Dykman (1.36217°S, 78.10865°W, 2,221 m), Cerro de Abitagua, Llanganates National Park on 24 December 2021.

Diagnosis. Pristimantis paganus sp. nov. (Figs. 2–7) is a member of the subgenus Cryptomantis and P. bicantus species group, characterized by the following combination of characters: (1) dorsal skin finely granular with dorso-laterally aligned flattened warts, areolate venter, discoidal fold present and visible posteriorly, dorsolateral folds weak and discontinuous in females, continuous in males; scapular W-shaped fold in males; (2) tympanic membrane and tympanic annulus present, round, horizontal diameter of tympanum equal to 38% of eye diameter, antero-dorsal margin with a supratympanic fold and large subconical postrictal tubercles; (3) snout short, subacuminate in dorsal view, rounded in profile with slightly flared lips; (4) upper eyelid with at least three to four large rounded tubercles surrounded by several small rounded tubercles, two subconical tubercles behind each eye; no cranial crest; (5) dentigerous processes of vomers, oblique in outline each process with 3–8 teeth; (6) vocal slits absent; no nuptial pad and no vocal sac; (7) Finger I shorter than Finger II; broad, expanded disks dilated with circummarginal grooves; (8) fingers with thin lateral cutaneous fringes; (9) subarticular tubercles, present, prominent; hyperdistal subarticular tubercle, present; ulnar tubercles absent; (10) heel with one subconical tubercle; outer edge with two flattened tubercles, tarsal fold absent; (11) inner metatarsal tubercle oval in females 3 times larger than rounded outer metatarsal tubercle; supernumerary tubercles absent; (12) toes with thin lateral fringes present, interdigital membrane absent, Toe V longer than Toe III; (13) The distinctive coloration of females, characterized by a uniformly black dorsum and black markings on a cream to pinkish-cream background on the flanks, belly, and throat, distinguishes this species from its congeners and other Pristimantis species inhabiting the eastern montane forests of Ecuador; (14) adult males, SVL = 17.20–21.5 mm (mean = 19.6, n = 13), females SVL= 24.8–29.8 mm (mean = 29.6 mm, n = 6), (Tables 2, 3, S3).

Table 2 Qualitative morphological traits.

Diagnostic characters used for comparison of Pristimantis paganus sp. nov. and its nominal sister species in the Cryptomantis subgenus.

Species	SVL (mm)	Snout in dorsal view	Snout in profile view	Disc shape	Dermal fold	Upper eyelid tubercles	Lateral cutaneous ridges on hands	Ulnar tubercles	Tubercles in heels	Tubercles on external border of the tarsus	Altitudinal range (m.a.s.l.)	
Female	Male	
P. paganus sp. nov.	24.84–29.8	17.20–21.5	Subacuminate	Rounded	Expanded	Defined and discontinuous dorsolateral dermal folds on the dorsum in females, defined continuous in males	Present, 3 to 4 rounded tubercles surrounded by several small ones.	Present	Absent	Subconical tubercle	Two flattened tubers	2,157–2,495	
P. bicantus	17.0–21.7	12.0–15.8	Rounded	Rounded	Slightly expanded	Absent	Present, low tubers	Absent	Absent	Absent	Absent	2,000–2,300	
P. burtoniorum	20.8–27.0	16.6–17.4	Subacuminate	Subacuminate	Expanded discs	Absent	Present, three to four large subconical tubers	Present	Small, conical	Small conical tuber	Present, round-shaped	2,940–2,970	
P. ganonotus	–	14.0–15.2	Acuminate	Protruding	Expanded	Absent	Absent	Absent	Indistinct	Nonconical	Absent	2,940–2,970	
P. marcoreyesi	22.8–30.8	14.3–22.2	Subacuminate	Angular	Slightly wider than the digits	Present, weakly defined	Present, two or more flattened to subconical tubercles present	Present	Small, flattened	Presents, rounded	Present, weakly defined	2,500–3,100	
P. nelsongalloi	12.0–17.0	18.5–21.7	Subacuminate	Rounded	Lanceolate discs	Dorsolateral folds thin anteriorly and discontinuous posteriorly	Present, four or five small supraoculars, flat and low	Absent	Present, small	One subconical and several small tubers	Absent	1,627–1,800	
P. prolatus	20.8–24.1	13.7–18.4	Acuminate	Truncated	Expanded	“H” shaped	Present, one conical	Absent	Absent	Present, conical	Present	1,140–1,933	
P. sacharuna	18.5–19.5	–	Subacuminate	Slightly rounded	Slightly wider than the digits, expanded, rounded	W-shaped scapular fold followed by fine scapular fold in shaped of an inverted “V”	Present, one subconical	Present	Present, subconical	Subconical tubercle	Two to 3 subconical tubers	2,200	
P. tungurahua	24.4–27.9	17.1–20.8	Subacuminate	Protuberant	Rounds, expanded	W-shape very evident in males, not very evident in females	Present, one or two subconical tubercles	Absent	Present, low	Present, conical	Present, conical	2,500–2,750	

Table 3 Morphometric measurements (millimeters) of the type series of Pristimantis paganus sp. nov.

Range of measurements (maximum, minimum, mean and ± standard deviation).

Measures	Males	Females	Juveniles	
n = 13	n = 7	n = 3	
Snout-vent length (SVL)	18.10–21.50	24.84–29.8	13.54–16.71	
(19.59 ± 1.33)	(28.18 ± 1.93)	(15.46 ± 1.37)	
Head width (HW)	6.52–7.97	9.52–11.80	5.61–6.33	
(7.6 ± 0.56)	(10.66 ± 0.76)	(5.96 ± 0.29)	
Head length (HL)	7.08–8.89	9.75–11.9	6.63–7.14	
(8.3 ± 0.56)	(10.97 ± 0.6)	(6.80 ± 0.23)	
Horizontal eye diameter (ED)	2.89–4.00	3.73–4.9	2.35–3.24	
(3.85 ± 0.21)	(4.23 ± 0.49)	(2.91 ± 0.40)	
Interorbital distance (IOD)	2.04–2.70	2.59–3.08	1.72–2.05	
(2.6 ± 0.14)	(3.32 ± 0.55)	(1.96 ± 0.17)	
Eye-nostril distance (EN)	1.8–2.69	3.2–3.85	1.54–1.90	
(2.2 ± 0.56)	(2.53 ± 0.40)	(1.77 ± 0.16)	
Eyelid width (EW)	1.76–3.00	1.92–3.20	1.26–1.47	
(2.35 ± 0.35)	(2.53 ± 0.40)	(1.34 ± 0.09)	
Tympanic diameter (TD)	0.84–1.70	1.45–2.30	1.84–0.93	
(1.70 ± 0.2)	(1.70 ± 0.26)	(0.89 ± 0.04)	
Tibia length (TL)	9.71–11.90	15.07–16.60	7.44–9.54	
(11.5 ± 0.56)	(15.72 ± 0.59)	(8.69 ± 0.90)	
Femur length (FL)	8.77–1.54	13.30–14.60	7.13–8.64	
(9.6 ± 0.84)	(13.89 ± 0.43)	(7.88 ± 0.61)	
Foot length (FoL)	8.23–10.77	13.38–14.7	6.76–7.87	
(9.20 ± 1.13)	(13.78 ± 0.43)	(7.43 ± 0.48)	
Hand length (HaL)	4.78–6.76	7.27–8.40	3.83–4.99	
(5.55 ± 0.49)	(8.09 ± 0.35)	(4.46 ± 0.47)	
Finger disc width III (F3D)	0.76–1.10	1.11–1.40	0.66–0.72	
(0.95 ± 0.21)	(1.20 ± 0.10)	(0.68 ± 0.02)	
Finger disc width IV (T4D)	0.51–0.98	0.87–1.00	0.57–0.63	
(0.8 ± 0.14)	(1.04 ± 1.10)	(0.60 ± 0.02)	

Comparison with other species (Fig. 2; Table 1). The distinctive coloration of females, with a uniformly black dorsum and black markings on a cream to pinkish-cream background on the flanks, belly, and throat, distinguishes this species from its congeners (P. bicantus, P. burtoniorum, P. ganonotus, P. marcoreyesi, P. nelsongalloi, P. prolatus, P. sacharuna and P. tungurahua) and other Pristimantis species inhabiting the eastern montane forests of Ecuador. In addition, dorsolateral dermal folds in P. paganus are weak and discontinuous in females, continuous with a W-shaped scapular fold in males (weak in males and females P. marcoreyesi; W-shape evident in males and not evident in females in P. tungurahua; thin anteriorly, discontinuous posteriorly in P. nelsongalloi; W-shaped scapular fold followed by fine scapular fold in shaped of an inverted “V” in P. sacharuna; absent in P. bicantus, P. burtoniorum, P. ganonotus; and H-shaped in P. prolatus).

Besides Pristimantis paganus is distinguished from P. marcoreyesi, P. tungurahua, P. nelsongalloi and P. sacharuna (characters of these species in parentheses) by lacking ulnar tubercles (small flattened in P. marcoreyesi; present low in P. Tungurahua; small in P. nelsongalloi; and subconical in P. sacharuna). It has rounded tubercles on upper eyelids (flattened to subconical in P. marcoreyesi; small and flattened in P. nelsongalloi; and subconical in P. sacharuna and P. tungurahua). The external metatarsal tubercle is rounded (subconical in P. marcoreyesi; round-subconical in P. nelsongalloi and P. sacharuna).

Likewise, juveniles males of the new species may be confused with those of P. bicantus, but can be distinguished by having fingers with thin lateral cutaneous fringes (absent in P. bicantus), heels with a subconical tubercle (absent in P. bicantus), and males lacking vocal slits (present in males from P. bicantus). Finally, P. paganus also differs from P. prolatus, P. ganonotus and P. burtoniorum by having a short, subacuminate snout in dorsal view (acuminate in P. prolatus and P. ganonotus; large and subacuminate in P. burtoniorum). The upper eyelid with rounded tubercles (one conical in P. prolatus; subconical in P. burtoniorum; and absent in P. ganonotus). Other differences are mentioned in Table S3.

Description of the holotype (Fig. 3). Adult gravid female. Head longer than wide; snout short, subacuminate in dorsal view, rounded in profile; lips flared, eye-nostril distance 12% of SVL, canthus rostralis straight; loreal region concave, protruding nares directed laterally; interorbital area flat, no interorbital fold, interorbital distance wider than upper eyelid, ~90%; no cranial crest; upper eyelid with at least three to four large rounded tubercles, tympanic annulus present, tympanic membrane differentiated from surrounding skin, tympanic ring evident surrounding 1/2 of tympanic membrane, upper margin covered by thick tympanic fold, diameter of tympanum equals 38% of eye diameter, underside of tympanum with large subconical postrictal tubercles; choanae small, oval in outline, not covered by palatal floor of maxilla; dentigerous processes of vomers present oblique in outline with 4 to 8 teeth, tongue as broad as long, slightly heart-shaped 40% attached to floor of mouth.

Skin of dorsum finely granular with flattened warts arranged dorso-laterally, weak and discontinuous dorsolateral folds; belly strongly areolate; discoidal fold present and visible posteriorly, cloaca granular, and the slender arms are free of tubercles on both the dorsal and ventral surfaces of the forearm. Ulnar tubercles absent; broad truncated disks on fingers II to IV, with circummarginal indentations, subarticular tubercles rounded and flattened in lateral view with thin lateral cutaneous fringes; thenar tubercle oval with heart-shaped palmar tubercle, palmar surface without supernumerary tubercles with scattered micro-granulations on hands, hind limbs slender, length of tibia equals 56% of SVL, no tubercle on outer edge of tibia, with a small subconical tubercle on heel, inner tarsal fold absent, all fingers with thin cutaneous ridges, without digital membranes; subarticular tubercles rounded and flattened in profile view; hyperdistal subarticular tubercle, present, elongated transversally; expanded disks on all toes, larger than those of the hand. Toe V longer than III, not extending beyond distal subarticular tubercle of Toe IV.

Measurements (in mm) of holotype. SVL = 27.23; HW = 10.24; HL = 11.21; ED = 3.92; IOD = 2.59; EN = 3.2; TD = 1.51; TL = 15.45; EW = 2.86; FL = 13.49; FoL = 13.54; HaL = 8.10; F3D = 1.18; T4D= 1.15

Holotype coloration in preserved (Fig. 3). Dorsal surface, forelimbs and hindlimbs homogeneously black with a thin creamy brown interorbital band. Inner edge of arm and forearm insertion, dorsal surfaces and bases of Toe I and II grayish brown, flanks marbled with grayish cream markings, ventrally with grayish cream coloration on belly, throat, thigh and tibia surfaces, belly on cream background mottled with black markings, throat on cream background and brown spots with large black markings, dorsal surfaces of thighs with black bands separated by light brown interspaces. Grayish cream line on the upper lip interspersed with black interspaces. Lower lip with black background separated by three grayish cream bands.

Holotype coloration in life (Fig. 5). Dorsal surface forelimbs and hindlimbs homogeneously black with a thin interorbital band of light pinkish brown, irregular pattern extending on the front margin of the eyelid to the tip of the snout. Inner edge of arm insertion, forearm, lateral dorsal surfaces, thighs and belly pinkish brown with black blotches, ventrally pinkish cream coloration on surfaces of belly, throat, thighs and tibia, belly and throat on grayish cream background mottled with large black markings and light brown blotches, dorsal surfaces of thighs with black bands separated by pinkish brown interspaces. Pinkish-brown line with small pinkish blotches more pronounced on the upper lip interspersed, discontinuous by black interspaces. Lower lip on black background separated by cream bands. The iris is bicolor golden with fine black reticulations in the upper part and a horizontal wide copper medial band.

Variation (Figs. 6, 7). Pristimantis paganus shows sexual dimorphism in body size and coloration. Males are smaller than females (Fig. S2; see Table 2 for details in morphometric measurements of the type series). In life (Fig. 7), females have a light grayish-brown coloration along the canthus rostralis, extending from the interorbital area of the head to the tip of the snout, while in adult males this area is orange-brown, and in juvenile males it is mustard-colored. An irregular blotch may or may not be present in the center of the canthus rostralis, being black in females and brown in males. Irregular marbled blotches are present on the ventral surface appearing black on a pinkish-cream background in females and brown on an orange-cream background in males. Dorsally, females exhibit a predominantly black coloration, while males show an orange-brown dorsum with dark brown W-shaped and inverted V-shaped scapular markings.

In ethanol (Fig. 6), the dorsal coloration pattern varies from a homogeneously black, light gray to grayish brown dorsum with a dark interorbital bar. The ventral coloration differs in the intensity and abundance of the botches, females possess abundant thick black spots, whereas in males it ranges from light brown, grayish brown to dark brown. The thighs, flanks and throat have a grayish cream coloration with large black spots in females and brown or grey in males.

Etymology. The specific epithet is from the Latin word “paganus”, an adjective derived from “pagus”, which refers to the inhabitants of the forest or village, far from civilization and towns, referring to the remote and unexplored sites where this species inhabits, in the montane cloud forests at the Colonso Chalupas Biological Reserve and Llanganates National Park, northeastern flanks of the Andes in Ecuador.

Distribution and natural history (Figs. 8, 9). Pristimantis paganus has been recorded from two localities separated by approximately 50 km, located in northeastern Ecuador in the provinces of Napo (Cordillera de Guacamayos = Guacamayos mountain range, Colonso Chalupas Biological Reserve) and Pastaza (Cerro de Abitagua, Llanganates National Park) (Fig. 8). These localities belong to the northern montane evergreen forest of the Eastern Cordillera of the Andes (MAE, 2013). The type locality and other collection sites are characterized by primary forest with abundant epiphytic plants (bromeliads, ferns, and moss), typical of the Amazonian cloud forest (Fig. 9). The type series of new species was collected at night between 22:00 and 01:00, on leaf litter, branches, bushes, and bromeliad leaves up to 120 cm above the ground. The holotype, an adult gravid female with eggs ready for oviposition, was recorded on the night of 21 November 2021.

Figure 8 Distribution of the subgenus Pristimantis (Cryptomantis) in eastern Ecuador (blue rectangle) based on the phylogenetic database.

(A) Subclade A, corresponding to localities of the Pristimantis prolatus species group; (B) subclade B, corresponding to localities of the Pristimantis bicantus species group; (C) localities of species from the subclade B, Pristimantis bicantus species group in the Pastaza valley (red rectangle in B). Major cities or towns in the eastern flanks of the Andes are marked as white circles with a dot. Map credit: Digital Elevation Model (DEM) developed by the Ecuadorian Government through the Ministry of Agriculture’s geospatial system, SIGTIERRAS, available at: http://geoportal.agricultura.gob.ec:8090/sinat_web_descarga/login.

Figure 9 Habitat and specimens of Pristimantis paganus sp. nov.

(A) Cloud forest with epiphytic plants, type locality, Colonso Chalupas Biological Reserve; (B) female, DHMECN 19962; (C) paratype, male, DHMECN 15602; (D) paratype, adult female, DHMECN 17222. Photographs: Jorge Brito (A), Keyko D. Loza-Carvajal (B), H. Mauricio Ortega-Andrade (C), Zane Libke (D).

Conservation status. During sampling in October 2016, July 2017 and November 2021, specimens of Pristimantis paganus were recorded infrequently (<25 individuals). However, this species is associated to primary forest in two protected areas Colonso Chalupas Biological Reserve and Llanganates National Park, on the northeastern flanks of the Amazonian Andes of Ecuador. Although viable populations (males, females and juveniles) of this species have been found within the National System of Protected Areas (SNAP), the region where it is distributed could be threatened by illegal mining, deforestation and climate change (Paz Cardona, 2022). According to the IUCN Red List criteria (IUCN, 2022), based on scarce number of localities and small distributional range with potential threats in the next decades, this species is proposed as “Near Threatened” (NT).

Discussion

Cryptomantis, a new subgenus of Pristimantis

We described a new subgenus for Pristimantis and a new Pristimantis species from the Colonso Chalupas Biological Reserve and Llanganates National Park, northeastern Ecuador, based on phylogenetic, genetic, morphological and geographic lines of evidence. The clade described as subgenus Cryptomantis, include two subclades described as the Pristimantis prolatus (three species) and the P. bicantus (six species) groups, shown strong phylogenetic support, confirming their monophyly (Fig. 1). The recognition of Cryptomantis, like other subgenera of Pristimantis (e.g., Hypodiction, Huicundomantis, Trachyphrynus), is justified by its well-supported monophyly, despite the absence of identified morphological synapomorphies (Hedges, Duellman & Heinicke, 2008; Ospina-Sarria & Grant, 2021). Padial, Grant & Frost (2014) reformulated the species series and species groups within Pristimantis, proposed by previous authors (i.e., Hedges, Duellman & Heinicke, 2008; Lynch & Duellman, 1997; Lynch & Duellman, 1980; Pinto-Sánchez et al., 2012), to make them explicitly monophyletic. However, nominal species related to the Cryptomantis subgenus (P. ganonotus, P. prolatus, P. tungurahua,) were unassigned to any of the taxonomic groups in Pristimantis (Padial, Grant & Frost, 2014). On the other hand, P. bicantus was described and assigned to the Pristimantis myersi species group (Guayasamin & Funk, 2009; Padial, Grant & Frost, 2014), but excluded from this group and leave as unassigned by Ortega, Brito & Ron (2022). Franco-Mena et al. (2023) recognized that Pristimantis bicantus, P. nelsongalloi, and P. sambalan form a well-supported clade nested within P. caprifer, suggesting that all of them should be assigned to the Pristimantis euphronides species group as defined by Targino (2016). Our results discard a close phylogenetic relationship between P. caprifer and P. euphronides + P. shrevei clade, and all other members of the subgenus Cryptomantis, in agreement with previously phylogenetic studies (Mendoza et al., 2015; Ortega-Andrade & Venegas, 2014; Padial, Grant & Frost, 2014; Waddell et al., 2018) (Fig. S1).

We observe two variant traits among species in the Cryptomantis clade. These include plump-bodied terrestrial frogs (mostly species of the Pristimantis bicantus species group), and slender-bodied bush frogs (distributed among the P. prolatus and P. bicantus species groups). The presence of dorsolateral dermal folds is a characteristic observed in both subclades of Cryptomantis; however, it is not exclusive to this group and also occurs in other groups within Pristimantis (Hedges, Duellman & Heinicke, 2008; Páez & Ron, 2019). We advocate for more comprehensive future analyses to determine whether diagnostic characters, such as dorsolateral dermal folds or other dermal structures (Guayasamin et al., 2015), are homologous or homoplastic among the Pristimantis clades (Bejarano-Muñoz et al., 2022; Ospina-Sarria & Grant, 2021; Ron et al., 2020).

Currently, three subgenera are recognized within the genus Pristimantis: Hypodictyon (Hedges, Duellman & Heinicke, 2008; Heinicke et al., 2018), Huicundomantis (Ortega, Brito & Ron, 2022; Páez & Ron, 2019) and Trachyphrynus (Franco-Mena et al., 2023). The description of the new subgenus Cryptomantis does not cause taxonomic instability due to its strong monophyletic support; instead, it provides an additional step toward classifying and subdividing Pristimantis into well-supported cladistic groups.

Cryptic diversity and the need for further taxonomic work

Species of the Cryptomantis subgenus are distributed in the eastern foothills of Ecuador. Speciation within this region is hypothesized to result from geographically restricted allopatry, with isolation occurring due to the uplift of the Cordillera Real Amazónica, the Abitagua and Guacamayos mountain ranges, and the occurrence of major Amazonian rivers such as the Napo and Pastaza (Fig. 9) (Duarte, 2013). A striking pattern of cryptic diversity and high species replacement in small areas in the Pastaza basin reveals a complex biogeographical and evolutive history for terrestrial frogs in the region (Reyes-Puig et al., 2022). The inclusion of topotypic sequences of Pristimantis prolatus (sensu stricto) and other Pristimantis species in our work, facilitated the understanding of species limits and the cryptic diversity of lineages, previously considered to have wide distributions in eastern Ecuador (Bejarano-Muñoz et al., 2022; Ortega, Brito & Ron, 2022; Reyes-Puig et al., 2022). For example, in contrast to the phylogeny of Ortega, Brito & Ron (2022), we determined that the topotyptic populations of Pristimantus prolatus are sister of the Pristimantis Candidate Sp.1, and Pristimantis bicantus are sister of Pristimantis Candidate Sp. 5 (See comments for definition of species groups; Fig. 1). The genetic sequence of Pristimantis prolatus (KU 177433), as used in Hedges, Duellman & Heinicke (2008) and Padial, Grant & Frost (2014), places this population near the type locality (Río El Salado). It shows strong support for nesting within the topotypic series provided here from Río Reventador. Like other Pristimantis speciation events in the upper Napo River basin (Bejarano-Muñoz et al., 2022), we identified cryptic diversity in Pristimantis prolatus complex, with at least one Unconfirmed Candidate Species (Fig. 1), by determining broad phylogenetic divergences with isolated populations from Guacamayos mountain range to Sangay National Park, compared with topotypic populations. The inclusion of the Pristimantis burtoniorum holotype sequence provides confidence about the phylogenetic relationships in the Pristimantis (Cryptomantis) prolatus species group, determining it as a monophyletic, well-supported clade (Fig. 1). Our evidence is robust to assign sensu stricto populations of Pristimantis bicantus and their phylogenetic relationships (Fig. 1; See comments on species group definition) within a diverse clade that also includes P. tungurahua, P. bicantus, P. sacharuna, P. nelsongalloi, P. marcoreyesi, P. paganus and four candidate species (P. c.sp. 2, P. c.sp. 3, P. c.sp. 4, P. c.sp. 5) in the Pristimantis (Cryptomantis) bicantus species group (Fig. 1). The new species, Pristimantis paganus, is endemic to the Guacamayos mountain range and Llanganates in the northeastern Andean foothills of Ecuador, 2157–2500 m a.s.l.

We highlight the need to include topotypic specimens when working with speciose groups, like Pristimantis, integrating several lines of evidence to test cryptic diversity hypotheses, to avoid underestimating species limits, promoting taxonomic instability and infer erroneous phylogenetic relationships. The recognition of vast cryptic diversity in Pristimantis in a geographically restricted region promote efforts to taxonomic studies and research toward unexplored areas (Ortega, Brito & Ron, 2022; Páez & Ron, 2019; Reyes-Puig et al., 2022; Reyes-Puig et al., 2014), like the Colonso Chalupas Biological Reserve and Llanganates National Park in Ecuador.

Conclusions

We described a new species, Pristimantis paganus, from the Colonso Chalupas Biological Reserve and Llanganates National Park, northeastern Ecuador. Furthermore, we define a new subgenus, Cryptomantis, within Pristimantis that includes the Pristimantis prolatus and Pristimantis bicantus species groups. Our results indicate at least five Candidate species within Cryptomantis. We emphasize the importance of including topotypic specimens to analyze and compare specios groups to delimiting species, like Pristimantis.

Supplemental Information

Supplemental Information 1 Maximum likelihood phylogenetic tree.

Phylogenetic relationships of the genus Pristimantis, with the position of the Cryptomatis subgenus.

Supplemental Information 2 Morphometric comparisons between males and females of Pristimantis paganus sp. nov.

(A) Principal component analysis, with Snout vent length (SVL), Tibia length (TL) and Femur length (FL) as variables which most explain data variance (PC1 and PC2=98%); ellipsoids represent 95% of confidence limits; (B) Correlogram of morphometric measurements with Linear r statistics represented as colored ellipsoids; Boxplot comparisons of (C) Snout vent length (SVL) and (D) Femur length (FL) of males and females; asterisk represent statistical differences inferred by Mann-Whitney U test.

Supplemental Information 3 Concatenated matrix with 1155 taxa terminals and 2981 genetic characters.

Partitions in the matrix: 16S = 1–796; 12S = 797–1682; COI = 1683–2348; RAG = 2349–2981.

Supplemental Information 4 Maximum likelihood tree (.tree).

Phylogenetic relationships of the genus Pristimantis, with the position of the Cryptomatis subgenus in *.tree format from IQTREE.

Supplemental Information 5 Concatenated matrix with 64 taxa terminals and 3371 genetic characters.

Partitions in the matrix: 16S = 1–1234; 12S = 1235–2090; COI = 2091–2741; RAG = 2742–3371.

Supplemental Information 6 Genbank accession numbers corresponding to species assigned to the subgenus Cryptomantis.

Sequences generated with Oxford Nanopore Technology sequencing.

Supplemental Information 7 Primers used to amplify the three mitochondrial and one nuclear gene in this study.

Universal Oxford Nanopore Technology adapters were attached to Forward (TTTCTGTTGGTGCTGATATTGC) and Reverse (ACTTGCCTGTCGCTCTATCTTC) primers.

Supplemental Information 8 Mean interspecific genetic p-distances (values below the diagonal) in the 16S rRNA gene between species of the subgenus Cryptomantis.

The minimum, maximum and standard deviation are shown above the diagonal.

Supplemental Information 9 Qualitative morphological characters.

Comparison of Pristimantis paganus sp. nov. and its nominal sister species in the Cryptomantis subgenus.

The authors would like to thank Andrea Carrera and Katherine Apunte for their support in the laboratory work on molecular data, and to Zane Libke for allowing the use of data and photographs from Llanganates National Park. We thank the Korea International Cooperation Agency (KOICA) for providing barcoding-producing capabilities through the National Institute of Biological Resources of Korea (NIBR) to authors KDLC, HMOA, WQ, and MYM.

Additional Information and Declarations

Competing Interests

The authors declare that they have no competing interests.

Author Contributions

Keyko D. Loza-Carvajal conceived and designed the experiments, performed the experiments, analyzed the data, prepared figures and/or tables, authored or reviewed drafts of the article, and approved the final draft.

Mario H. Yánez-Muñoz conceived and designed the experiments, performed the experiments, analyzed the data, prepared figures and/or tables, authored or reviewed drafts of the article, and approved the final draft.

Walter Quilumbaquin performed the experiments, analyzed the data, authored or reviewed drafts of the article, and approved the final draft.

H. Mauricio Ortega-Andrade conceived and designed the experiments, performed the experiments, analyzed the data, prepared figures and/or tables, authored or reviewed drafts of the article, and approved the final draft.

Field Study Permissions

The following information was supplied relating to field study approvals (i.e., approving body and any reference numbers):

Specimens and tissue samples obtained following technical protocols proposed by Heyer et al. (1994). Type specimens were deposited at the Instituto Nacional de Biodiversidad INABIO, Quito, Ecuador (DHMECN), under permits MAE-DNB-CM-2016-0045, 2017-0062 and 2019-0120 issued by the Ministry of Environment, Water and Ecological Transition of Ecuador.

DNA Deposition

The following information was supplied regarding the deposition of DNA sequences:

The 62 vouchers and sequences at Genbank are available in Table S1.

Data Availability

The following information was supplied regarding data availability:

The raw data are available in the Supplemental Files.

New Species Registration

The following information was supplied regarding the registration of a newly described species:

Publication LSID: urn:lsid:zoobank.org:pub:11445DCE-6CC1-4BA6-9ACC-1B070E79AA24

Pristimantis (Cryptomantis) subgenus LSID: urn:lsid:zoobank.org:act:AD83BF8F-72C0-4453-ABA8-705C2F258944

Pristimantis paganus species LSID: urn:lsid:zoobank.org:act:D532641A-414E-4580-A506-FDAEFDA34B0E.

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
