# Peer review of "Vast cryptic diversity in direct-developing frogs Pristimantis (Anura: Strabomantidae): a new subgenus and the description of a new species from the eastern Andes of Ecuador"

_PeerJ, doi:10.7717/peerj.20512_

## Round 0.1 · original submission · Major Revisions

· Academic Editor

Major Revisions

Dear Dr. Ortega-Andrade, I ask you to carefully improve the manuscript in accordance with each of the reviewers' comments. I hope that the new version of your article will be approved for publication as soon as possible.

·

Basic reporting

The authors describe a new species of Pristimantis from Andean Ecuador and define a new subgenus. The morphological and genetic methods they applied are correct, and their conclusions are justified.
I provided numerous comments/suggestions for improving sentence structure, marking errors, or clarifying unclear statements. Avoid unnecessary filler words, such as adding "frogs" or "genus" to Pristimantis, which is trivial knowledge.
Keep your species diagnosis short and do not include variation for males and females except for different SVL or male sex characters. Variation is very important and should be listed under "Variation".

Experimental design

I would add to the methods/materials how eggs (the holotype is gravid!) were measured, and provide measurements of eggs with average and sample size to Natural History.

Validity of the findings

Description of a new species and definition of a new subgenus are justified.

Additional comments

I recommend shortening your manuscript title by deleting trivial knowledge about the vast cryptic diversity in Pristimantis. Instead, focus on your new species and the new subgenus.

Your images are overall very good and detailed, but the incorrect/incomplete figure legends are very annoying. Use a consistent style for letters and add letters to the figures (some have letters, others do not, and others are missing letters).

In your holotype description, you need to refer to each figure and the figure letter for the characters described. The reader does not want to search for them.

Reviewer 2 ·

Basic reporting

The manuscript is generally clear and well-structured, and the figures and tables are informative. Citations are appropriate. Figures and raw data are provided, although some figures could be simplified for clarity (see Additional Comments).

Experimental design

The methodological framework is solid, integrating morphological, molecular, and phylogenetic analyses. Ethical standards and permits are mentioned. However, some technical details should be clarified:
- Number of specimens used in morphological analyses.
- A full list of revised specimens should be included in the supplementary material.

Validity of the findings

The phylogenetic analyses are robust, and the results justify both the new species and the proposed subgenus. The new species is well diagnosed morphologically and genetically. The subgenus is strongly supported molecularly, although no unique morphological synapomorphies are identified. The authors should strengthen the diagnosis by emphasizing combinations of characters (even if not unique).

Additional comments

This manuscript is clear and well written, and it represents a valuable contribution to the knowledge of the amphibian fauna of the northeastern Andes of Ecuador. I have only minor comments:

Figures: Figure 8 is overloaded. I suggest simplifying the PCA and boxplot panels to highlight sexual dimorphism and moving the correlogram to the supplementary material. However, I do not consider this figure indispensable for validating the new species, since the information is already included in the morphometric tables and can be described in the text with its statistical support. Do all species in the group show the same dimorphism pattern, or is it particular to P. paganus? If Figure 8 is retained, I suggest including a discussion of sexual dimorphism in comparison with other species of the group.
In addition, why was a linear morphometric analysis not performed among the species closely related to P. paganus?

Figure 9: I suggest highlighting P. paganus sp. nov. with a distinctive symbol, and representing subclade A with a uniform shape (e.g., squares). It may also be clearer to present all populations in a single map.

Tables: It would be useful to include a table with genetic distances between the new species and its closest relatives.

Supplementary material: Please check that each title corresponds to the correct material (e.g., Stable2 and Stable_3).

Annotated reviews are not available for download in order to protect the identity of reviewers who chose to remain anonymous.

---

## Round 0.2 · accepted · Accept

· Academic Editor

Accept

Dear Dr. Ortega-Andrade, I congratulate you on the acceptance of this article for publication.

·

Basic reporting

The authors revised their manuscript carefully and in detail. My previous concerns have been addressed, and the manuscript is in excellent shape, supported by high-quality images. Their scientific conclusions, the description of a new species of Pristimantis, and the proposal of a new subgenus are justified.

Experimental design

All excellent!

Validity of the findings

All justified!

Additional comments

I found one tiny error in the abstract, L25: change to read "group includes".

Reviewer 2 ·

Basic reporting

no comment

Experimental design

no comment

Validity of the findings

This manuscript provides sufficient evidence to support the distinctiveness of a new species of Pristimantis and the establishment of Cryptomantis subgen. nov. is sound and consistent with recent taxonomic practice in Pristimantis. The authors provide sufficient information about the study system and cite relevant studies. The methods were described with sufficient detail, and the authors followed general standards in amphibian taxonomy.

Additional comments

no comment

Reviewer 3 ·

Basic reporting

The manuscript is written in clear, professional English and follows the standard scientific format (Abstract, Introduction, Materials & Methods, Results, Discussion, Conclusions). The overall presentation is coherent and easy to follow. Figures and tables are relevant, of high quality, and effectively support the text.

Experimental design

The study is well-designed and methodologically rigorous. It fits well within the aims and scope of PeerJ and presents original primary research.
The authors clearly define the research objectives and the taxonomic problem they address — the identification of cryptic diversity within Pristimantis and the justification for a new subgenus and species.
Sampling design is appropriate (2016–2021) and includes detailed locality data, coordinates, and permits (MAE-DNB-CM-2016–0045, 2017–0062, 2019–0120).
Methods for molecular work (Sanger and Oxford Nanopore sequencing) are robust and described in detail, allowing reproducibility.
The analyses (MAFFT, IQTree, ModelFinder, Mesquite, PAST) are state-of-the-art and correctly applied.
Ethical and legal requirements were met, with voucher specimens deposited in INABIO (DHMECN collection).
The description of the Cryptomantis subgenus follows ICZN standards and includes valid ZooBank registration (urn:lsid:zoobank.org:pub:11445DCE-6CC1-4BA6-9ACC-1B070E79AA24).
Overall, the study design meets high technical and ethical standards.

Validity of the findings

The data are robust, statistically sound, and properly controlled. Phylogenetic reconstructions show strong support (bootstrap 98–100%) for both the new subgenus (Cryptomantis) and the new species (Pristimantis paganus).
Uncorrected 16S p-distances between P. paganus and related taxa (2.4–7.5%) clearly justify species-level differentiation.
Morphological and molecular evidence are presented in an integrative framework, in line with modern taxonomic standards.
The conclusions are well-supported by the data and appropriately linked to the research question.
The work provides a valuable contribution to Neotropical herpetology and integrative taxonomy, clarifying long-standing uncertainties in Pristimantis systematics.

Additional comments

The manuscript is scientifically sound and presents a significant addition to our understanding of Andean biodiversity. I commend the authors for combining traditional morphology with advanced genomic methods and for including topotypic specimens, which greatly strengthen their conclusions.